palaeontology, evolution

early bursts, disparity, GOBE

**Author for correspondence:**
Curtis R. Congreve
e-mail: crcongre@ncsu.edu

# An early burst in brachiopod evolution corresponding with significant climatic shifts during the Great Ordovician Biodiversification Event

Curtis R. Congreve[1], Mark E. Patzkowsky[2] and Peter J. Wagner[3]

[1]Marine, Earth and Atmospheric Sciences, North Carolina State University, Jordan Hall, Raleigh, NC 27607, USA
[2]Department of Geosciences, Pennsylvania State University, Deike Building, University Park, PA 16802, USA
[3]Department of Earth and Atmospheric Sciences and School of Biological Sciences, University of Nebraska, Lincoln, NB 68588, USA

CRC, 0000-0003-3476-1257; MEP, 0000-0002-1761-3298; PJW, 0000-0002-9083-9787

We employ modified tip-dating methods to date divergence times within the Strophomenoidea, one of the most abundant and species-rich brachiopod clades to radiate during the Great Ordovician Biodiversification Event (GOBE), to determine if significant environmental changes at this time correlate with the diversification of the clade. Models using origination, extinction and sampling rates to estimate prior probabilities of divergence times strongly support both high rates of anatomical change per million years and rapid divergences shortly before the clade first appears in the fossil record. These divergence times indicate much higher rates of cladogenesis than are typical of brachiopods during this interval. The correspondence of high speciation rates and high anatomical disparity suggests punctuated (speciational) change drove the high frequencies of early anatomical change, which in turn suggests increased ecological opportunities rather than shifting developmental constraints account for high rates of anatomical change. The pulse of rapid evolution began coincident with cooling temperatures, the start of major oscillations in sea level and increased levels of atmospheric oxygen. Our results suggest that these factors permitted major geographical and ecological expansion of strophomenoids with intervals of geographical isolation, resulting in elevated speciation rates and corresponding elevated frequencies of punctuated change.

## 1. Introduction

Determining the relationship between global environmental changes and the diversification of life presents two especially difficult challenges for earth scientists and biologists. The first challenge is to constrain the absolute timespan between an environmental shift and the radiation of diversity and disparity in a clade [1]. The second challenge is to identify evolutionary modes that explain both rates of anatomical change and rates of divergences (speciation) caused by the environmental shift [2]. New approaches that combine tip-dating methods and birth–death-sampling models [3–5] can address both of these challenges. These approaches are especially important for untangling the causal relationships during major evolutionary radiations, in which multiple organismal groups diversify and for which multiple possible causal factors exist.

The Ordovician radiations offer multiple potential examples of environmentally driven radiations. The Ordovician witnessed a fourfold increase in generic richness relative to the preceding Cambrian [6–8], including the ascendancy of the Palaeozoic Fauna [9], a major expansion of ecological roles [10–14] and rapid increases in morphological disparity in many clades [15–21]. During this

interval, several major environmental changes occurred, each of which might have driven some or all of the Ordovician radiations [1]. These include increased planktic productivity starting in the Late Cambrian [22], increased oxygenation starting at the onset of the Middle Ordovician [23], long-term global cooling throughout the Ordovician with glaciation beginning partway through the Middle Ordovician [24], major oscillations in sea level [24] and elevated orogenic activity beginning in the late Middle Ordovician [25].

In recent years, palaeontologists [7,26–28] have devoted considerable effort towards assessing the timing of Ordovician radiations among whole faunas, particularly during the Great Ordovician Biodiversification Event (GOBE) in the Middle Ordovician (approx. 470–458 Ma). Less attention has been given to the diversification dynamics of particular clades with regard to the GOBE. Here, we examine a prominent participant of the GOBE, the Strophomenoidea (Brachiopoda), a brachiopod clade that appeared and rapidly diversified in the Middle Ordovician [29] while achieving nearly global distribution [30] and that also showed early peak morphological disparity (electronic supplementary material, figures S1 and S2). Using previously published phylogenetic relationships and character data [31,32], we employ tip-dating methods [33,34] plus prior probabilities on branch durations and divergence times using a fossilized-birth–death (FBD) model [3,4] to find not just the best-supported divergence times, but also the best-supported models of how rates of anatomical change and cladogenesis changed over time. We then contrast the best-supported evolutionary model with the potential environmental drivers of Ordovician radiations outlined above in order to determine both the likely environmental triggers for evolutionary radiation in this clade, as well as the primary mode of evolution at this time period.

## 2. Material and methods

### (a) Character data and phylogeny

We used the topology and character matrix from a previously published strophomenoid phylogeny to obtain both evolutionary relationships and 65 shell characters for 39 Ordovician strophomenoid species [31,32]. These 39 species were used as exemplars [35] for 38 genera and represented an even sampling across the four major strophomenoid families during this time period and sampled all of the major biogeographic units from the Ordovician [32] (see also electronic supplementary material, figure S8). Because the genus *Leptaena* might be paraphyletic to other members of the Rafinesquinidae, we treated the type species (*L. rugosa* Dalman 1828) as the exemplar for the genus and the second species (*L. richmondensis* Foerste 1909) as a monotypic genus. We used the published cladogram [31] as a model phylogeny in this analysis.

### (b) Stratigraphic data

Stratigraphic data serve two roles in our analyses. First, they provide the ranges of the 38 genera that we analyse directly. Second, these data for Cambrian and Early Ordovician brachiopods provide a basis for assessing origination, extinction and sampling rates before those 38 genera appear in the fossil record; these rates, in turn, are critical for assessing the possibility that strophomenoids began to diversify long before they appear in the fossil record, and thus are critical

for assessing null hypotheses of constant rates of diversification and anatomical change.

To assess stratigraphic preservation and trends in extinction and speciation across the Cambrian and Ordovician, we analysed the stratigraphic ranges of 12 284 strophomenide and other brachiopod species-level occurrences in the Paleobiology Database that come from 6928 Cambrian and Ordovician collections and 929 published sources downloaded on 9 August 2019 (electronic supplementary material, figure S7). We added over 2700 occurrences and 1100 collections to the Paleobiology Database for this study, in part to make sure that the oldest representatives of all analysed genera were included, but also to overcome possible geographical biases in previously entered data. We also updated stratigraphic data in another 250 existing collections. Because we derived genus ranges from species records and because many strophomenide genera include species that previously were classified in other genera, we entered nearly 660 species records and an additional 1200+ taxonomic opinions. We paid particular attention to strophomenide species with similar species epithets because these were the most plausible cases for representing the same species with 2+ genus opinions.

Because the Paleobiology Database returns only very general ages for collections, we used an external database built by one of us to provide the most exact dates possible based on conodont, graptolite, trilobite and chitinozoan zones [36]. We used this database to provide lower and upper bounds on possible first appearance dates for the analysed genera. We also used these data to generate origination, extinction and sampling rates for FBD priors on divergence times (see below).

Although our analyses were done at the genus level, the FBD analyses required that we used species-level origination, extinction and sampling rates. Although species have higher rates of origination and extinction than genera, species also have lower sampling rates: only one of (often) several species need to be sampled to find a genus. However, in order to accurately assess the quality of our stratigraphic record, we need the probabilities of missing any earlier strophomenoid species, not just those strophomenoid species that would be placed in a new genus by systematists [37]. In theory, the oldest sampled strophomenoids could range back to the Cambrian or earlier. We therefore require species-level rates to assess these probabilities.

### (c) Bayesian tip-dating with fossilized-birth–death priors

We compare three models of evolution to determine which one best fits the available data. The first model (Strict Clock + First Appearance [FA] Priors) assumes a constant rate of anatomical change sampled in the phylogeny, rates of origination, extinction and sampling matching those estimated from brachiopod occurrence data (electronic supplementary material, figures S9–S11) while allowing for uncertainty in true first appearance times of taxa [38] (electronic supplementary material, figure S2). Note that 'strict clock' here is essentially equivalent to phyletic gradualism or continuous anagenetic change along branches where expected change is a product of only a constant rate ($\alpha$) and time ($t$). The second model (Strict Clock + FA Priors + Branch Priors) assumes a constant rate of anatomical change and speciation while using the FBD method to estimate prior probabilities of

phylogenetic branch durations and divergence times as well as uncertain first appearances. The third model (Early Burst + FA Priors + Branch Priors) expands the second model by allowing elevated rates of anatomical change per million years early in the clade history (i.e. a 'big bang' or 'early burst' model [39,40]). This is still equivalent to a phyletic gradualism or continuous change model, but one in which expected change is given by $\int \alpha t$ and $\alpha$ shifts over time [41].

We use Lewis's Mk model [42] to calculate the likelihoods of rates and divergence times given anatomical data. For both strict clock and early burst analyses, we used a lognormal distribution to model rate variation among characters, as this fit the distribution of changes implied by parsimony slightly better than did a gamma distribution (electronic supplementary material, table S1). Under both strict clock and early burst models, the rate and divergence likelihoods represent the average likelihoods given four quartiles from a lognormal with median rates $\alpha_{Strict}$, $\alpha_{Early}$ or $\alpha_{Late}$ [43]. Finally, the best-fit lognormal implied at least two invariant characters, which matched the number of invariant characters from our character matrix. Thus, we did not adjust for ascertainment bias (i.e. characters that could have changed but did not) [42].

Prior probabilities of divergence times reflect origination, extinction and sampling [4]. All three parameters varied substantially over time among early Palaeozoic brachiopods [25] and this variation strongly affected divergence time priors [44]. Although 'skyline' models used in Bayesian phylogenetic studies allow us to find the most probable set of diversification and sampling rates given a tree, character data and rate model [45], these approaches have little power to resolve rate shifts preceding a clade's appearance in the fossil record. Instead, we based origination, extinction and sampling rates on the Paleobiology Database occurrences described above. We used a modified version of the three-timer method [46], but with uniform among-taxa sampling rates replaced with best-fit lognormal distributions of sampling rates given occurrence data for those intervals (electronic supplementary material, figures S4–S6). We used the database described above to assign collections to stage-slices [7,47]. We could not rely solely on strophomenoids because we needed to estimate prior probabilities of divergence dates that preceded the oldest known strophomenoid fossils. Therefore, we used the broader clade Rhynchonelliformea that includes strophomenoids (and their ancestors) to infer the Early Ordovician and Dapingian rates for the FBD analysis. Similarly, we used all brachiopods to generate Cambrian rates for our analyses. We used an R program written by one of us to determine the best rates and divergence times given basal divergence times from 466 to 521 Ma (electronic supplementary material).

Assigned first appearance dates can strongly affect the likelihoods and probabilities of rates and divergence times on a phylogeny [38]. We therefore repeated the analyses 100 times using different possible first appearances for each taxon. For any given genus, the probability of the 'true' first appearance was given by a β distribution with shape1 = 1 and shape2 = N, with N = the number of collections that *might* be first appearances. If N = 1, this is uniform; if N = 2, then this is a symmetrical curve around the median age. We did not use this distribution to initially model replicate analyses because 11 of the 203 collections that *might* represent a first occurrence for a genus might also represent a first occurrence for another genus. Therefore, we randomly assigned

dates to the 203 PBDB collections that might be the first appearance of a genus and then tallied the first appearances for all of the genera. The analyses described above were then executed. We then used the β distributions for each genus to put prior probabilities on the set of first appearances used in each replicate being the 'true' set. Note that we used these priors in all analyses; thus, we contrasted three models of evolution: Strict Clock + FA Priors, Strict Clock + FA Priors + Branch Priors and Early Burst + FA Priors + Branch Priors.

## (d) Reconstructed cladogenesis rates on the model tree

We analysed cladogenesis rates on trees using the same general approach that we used to put prior probabilities on branch durations. The sole difference was that we replaced the empirically estimated species-level origination rate when estimating the probability of sampling clades of any size with twice the hypothesized one. (This reflects an average of two species per strophomenoid genus.) Because branch durations typically were set by divergences between sister taxa rather than sampled ancestors, the likelihood of a cladogenesis rate given any one branch is the probability of one sampled cladogenetic event over that branch duration (with the probability of a cladogenetic event being sampled reflecting hypothesized origination rates and empirically estimated extinction and sampling rates) times the probability of zero sampled ancestors from that branch over that time. We did this for both constant cladogenesis and for shifting rates with declining rates of cladogenesis over time.

## 3. Results

We find very strong support for the Early Burst model suggesting that the strophomenoids included in this study diverged 3–4 Myr before the oldest taxa analysed here first appear in the fossil record and nearly coinciding with the oldest unambiguously identified strophomenoid species (*Hesperinia sinensis* Rong and Zhan 1999) in the fossil record [48] (figure 1; also electronic supplementary material, figures S7 and S8 [45]). The Strict Clock + FA Priors suggests that the Strophomenoidea diverged from the rest of the Strophomenida near the onset of the Ordovician more than 20 Myr before the strophomenoids made their first appearance in the fossil record (figure 2a and table 1). The Strict Clock + FA Priors + Branch Priors pushes the divergence halfway towards the first appearance of strophomenoids in the fossil record. The effect of FBD priors on branch durations alone is strong: Bayes Factors (BF) given Strict Clock + FA Priors + Branch Priors offer very strong support (BF = 85.7; figure 2b; note that this is based on the posteriors from the Strict Clock + FA Priors + Branch Priors model given the divergence dates favoured by the Strict Clock + FA Priors). The Early Burst + FA Priors + Branch Priors model suggests a divergence of 468 Ma (Dapingian) with credible intervals spanning the Dapingian. Bayes Factors indicate strong support for this model over the Strict Clock + FA Priors + Branch Priors model (BF = 20.4). This corresponds to a model in which rates of change prior to 462 Ma are 3–3.5 times greater than rates after 462 Ma.

The branch durations supported by the Early Burst + FA Priors + Branch Priors model indicates that the strophomenoids condensed considerable cladogenetic activity into a short time after the clade diverged from the rest of the

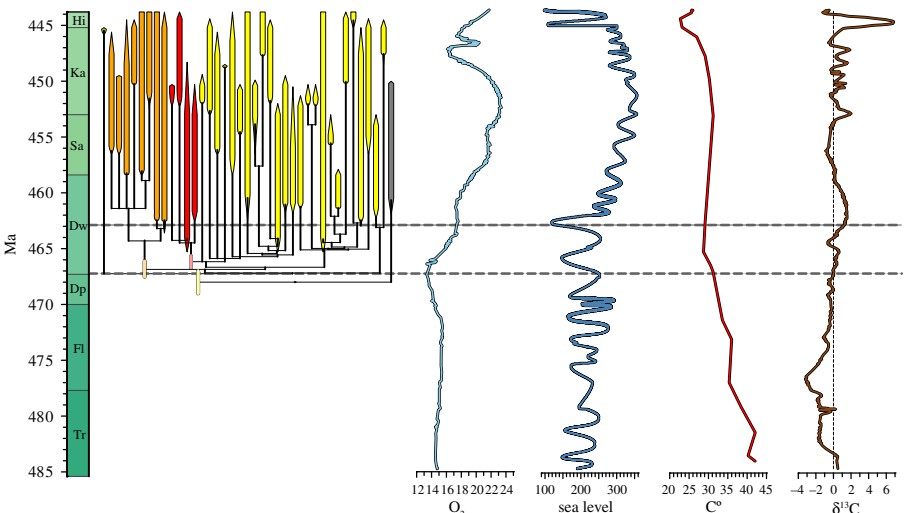

**Figure 1.** Divergence times by best Early Burst + FA Priors + FBD Priors model and proxies for Ordovician climate and environmental change. Dark red denotes members of the Glyptomenidae. Orange represents members of the Rafinesquinidae. Light yellow denotes members of the Strophomenidae. Pale bars give credible intervals for the divergences of the families from other brachiopods. 'Tails' at onset and end of ranges give the probability that the first appearance of a genus has happened by that time given fossil finds. Dashed line denotes the window in the first half of the Darriwillian into which much of strophomenoid evolution is compressed. Time scale and $\delta^{13}$C curve modified from [49]; atmospheric oxygen curve modified from [23]; temperature curve modified from [50]. See also [6]. (Online version in colour.)

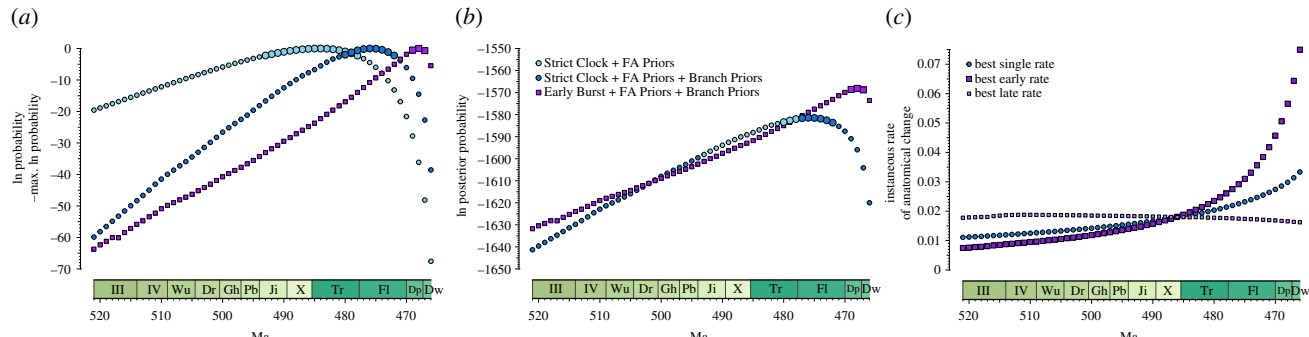

**Figure 2.** Divergences and character rates. (*a*) Probabilities of divergence times for all three models rescaled to maximum probabilities. Large symbols denote divergences within credible intervals (i.e. 95% of total probability). (*b*) Posterior probabilities of Strict Clock and Early Burst models given FBD priors on Branch Durations, rescaled to the most probable model. The credible intervals on the Strict Clock without Branch Priors are shown in light blue for contrast. The differences between the Strict Clock models and the Early Burst Models in the Late Cambrian and earliest Ordovician reflect high prior probabilities of first appearances for the Strict Clock example. (*c*) Most probable rates of change. These intersect at the most probable divergence time given the Strict Clock + FA priors as the early and late rates become identical to each other and the Strict Clock rate at this point. (Online version in colour.)

**Table 1.** Summary of the most probable trees given the three models considered here. Basal divergence indicates the most probable date at which the Strophomenoidea diverged from the rest of the Strophomenida. Numbers in brackets give 95% credible intervals. InP FAs gives log probabilities of first appearances. lnL $\alpha$ give the log-likelihood of the character rate hypothesis given divergence times. InP FBD gives the log-probability of failing to sample an ancestor or to sample another sister-taxon along some branch duration. In posterior gives the log of the posterior probability of divergence time and rate models.

| model | basal divergence | lnP FAs | lnL $\alpha$ | lnP FBD | ln posterior |
|---|---|---|---|---|---|
| Strict Clock + FA Priors | −484 [493, 478] | −71.48 | −1448.82 | −66.64 | −1586.95 |
| Strict Clock + FA Priors + FBD Priors | −476 [480, 472] | −71.48 | −1453.27 | −56.67 | −1581.43 |
| Early Burst + FA Priors + FBD Priors | −468 [467, 470] | −75.03 | −1456.29 | −36.78 | −1568.10 |

Strophomenida (figure 3). The best single rate of cladogenesis is within expectations given overall rates of species-level origination rates for the Rhynchonelliformea (electronic supplementary material, figure S10) even after allowing for multiple speciations per genus-origination [44]. However, many of the Dapingian and early Darriwilian branch durations best-fit high local rates of cladogenesis (figure 3). A simple exponential decrease in cladogenesis rates is significantly more likely than the best single rate (lnL = −100.6 versus lnL = −117.3; $p = 7 \times 10^{-9}$).

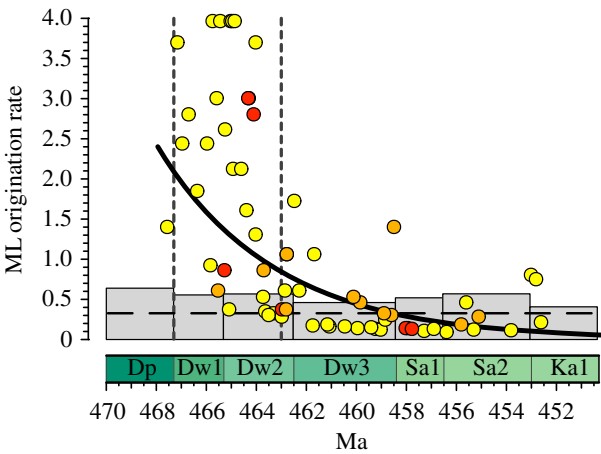

**Figure 3.** Rates of cladogenesis implied by the most probable Early Burst + FA Priors + Branch Priors model. Grey bars give typical turnover rates for given occurrence data for rhynchonelliform brachiopods (including strophomenoids). Dashed line gives the most likely single rate of cladogenesis. Curved line represents a best-fit exponential decay in origination rates. Dots represent origination rates for individual branches that would maximize the probability of the branch duration given sampling and extinction rates. Therefore, short branch durations generally elevate the likelihood of high rates of origination. (Note that high sampling and/or low extinction also encourage short branch durations.) Ages reflect the mid-points of the individual branches shown in figure 1. Colours and dashed lines as in figure 1. (Online version in colour.)

## 4. Discussion and conclusion

Taken together, our results indicate rapid morphological and phylogenetic diversification of strophomenoids concentrated in the first half of the Darriwilian. The early Darriwilian was also a time of significant changes in Earth's environment suggesting these changes may have caused the increased strophomenoid diversification (figure 1) [22]. A positive excursion in $\delta^{13}$C of carbonates occurred in the middle to late Darriwilian after the most rapid diversification. Thus, perturbations to the carbon cycle, such as increased productivity or an increase in the rate of organic carbon burial, were probably not causal factors of increased diversification. Instead, the onset of the clade's radiation coincided with the beginning of a long-term increase in atmospheric oxygen and at a time when average sea surface temperatures had decreased to modern levels and were rapidly declining. The subsequent interval over which strophomenoids radiated also coincided with two of the largest prolonged fluctuations in sea level thought to have occurred in the Ordovician. These environmental shifts need not be independent: decreasing temperatures permit increased dissolved oxygen in the oceans and can also induce glaciation that cause sea-level oscillations and change circulation patterns that can provide increased oxygen to ocean depths.

These large scale oscillations in sea level during the latest Dapingian and early Darriwilian [51] might have been particularly important, as sea level directly affects geographical barriers that promote vicariant and allopatric speciation [52] of the sort that would generate punctuated character change [53]. In fact, previous works on the other time periods [54] as well other radiations during the Ordovician [55,56] have suggested sea-level oscillation could function as a potential motor of speciation throughout deep time. This model also is consistent with Ordovician patterns of β-diversity [57]. Meanwhile, increasing oxygen concentrations and cooler temperatures may have

facilitated range expansion of existing taxa to previously marginal environments that were too warm or oxygen-poor to inhabit, which in turn could have increased the likelihood of speciation by dispersal and isolation of populations along the margins of the species range. For example, the Strophomenata might have lived infaunally [58] and increasing oxygen concentrations at the sediment–water interface could have provided an opportunity for the strophomenoids to diversify. It is also possible that the observed pulse in speciation could be the result of a combination of these environmental shifts. Changing sea level, global cooling and increased oxygenation may have synergistically acted to increase environmental heterogeneity in ocean environments during this time period, which in turn would increase the rates of allopatric speciation (synergistic environmental changes has been previously discussed in [2]).

The early burst model for the Strophomenoidea supports the idea of a distinct diversification event (GOBE) in the Middle Ordovician [2], although it suggests that the interval of rapid diversification for this clade could be shifted back to the Dapingian or early Darriwilian approximately 3–4 Myr before the first appearances in the fossil record of the members of the clade that we analyse (figure 1). On the one hand, this result represents another case in which the implications of taxonomic approaches are largely consistent with those of tree-based approaches [59]. On the other hand, our results also suggest that the patterns implied by traditional first–last appearance data [30] or capture–mark–recapture assessments using occurrence data [7,26,27] *understate* the pace of strophomenoid diversification. This is important because there are multiple major environmental shifts occurring during a relatively short period of time during the Ordovician and changes in just a few million years in the estimated onset of diversification have a profound effect on our understanding of the relevant drivers of strophomenoid evolution. The early burst model puts the main interval of diversification just when global temperatures decreased to modern values, at the onset of increased fluctuations of sea level, and essentially coincident with the beginning of a long-term increase in atmospheric oxygen (figure 3; lower dashed line). The close association of increased speciation with decreasing global temperatures, increased sea-level fluctuations and an increase in oxygen concentrations suggests that these environmental shifts may have facilitated ecological opportunities for increased speciation in the strophomenoid brachiopods and that the evolutionary response was rapid, possibly within a few hundred thousand years. Alternatively, a direct reading of the fossil record would put the main phase of diversification about 4 Myr later in the middle Darriwilian (figure 1; upper dashed line). This interpretation would demand a several million year lag in evolutionary response to environmental shifts if global cooling, sea-level fluctuations and increasing oxygen concentrations were the main drivers for diversification.

While the events of the GOBE resulted in increased rates of evolutionary change across nearly all animal orders, these shifts were not uniformly distributed across the tree of life [30]. Given that strophomenoids along with the closely related (and probably paraphyletic [31]) Plectambonitoidea diversified at appreciably greater rates than did other brachiopods at this time [30,52] (see also electronic supplementary material, figure S13), the pattern strongly suggests that strophomenoids, for whatever clade-specific reasons, took advantage of these environmental shifts in ways many other brachiopod clades

did not. In fact, the dramatic difference in rates observed in this study is likely to be a conservative estimate, given that our analyses assumed that strophomenoids and other brachiopods shared the same rates when we estimated branch priors. It is possible that strophomenoids were better able to take advantage of these environmental shifts due either to currently unknown differences in dispersal ability (allowing for greater instances of peripheral isolation due to shifting sea levels) or an ability to take advantage of greater ranges of oxygen at depth. Further phylogenetic tip-dating studies of other non-brachiopod taxa from this time period are necessary to determine if the observed pattern of evolutionary radiation in response to these environmental shifts is a more universal pattern among other invertebrate groups, or if it is highly clade specific.

There are two general and non-exclusive explanations for high rates of anatomical change early in clade history: low developmental constraints [60] and novel ecological opportunity [61]. High branching rates among genera coinciding with high rates of change offers two possible explanations in which one rate is an artefact of the other. Because taxonomists recognize new genera based on anatomical distinctiveness, high rates of anatomical change coupled with constant rates of speciation would result in more new genera per total descendants and generate the patterns we document [34,37]. This also predicts that the numbers of species per genus and the average numbers of species per genus over genus ranges should increase over time. This is not observed (electronic supplementary material, figure S10), suggesting that amounts of anatomical evolution per species did not change markedly over the Ordovician. This in turn supports novel ecological opportunity as the primary mode facilitating anatomical change rather than reduced developmental constraints.

The link between origination rates and frequencies of anatomical change offers further support for novel ecological opportunities as the primary driving force of the radiation when we consider alternative modes of speciation and anatomical change. The Mk model that we used in our analyses (see Material and methods) assumed continuous anatomical change (i.e. 'phyletic gradualism' [62]), with some instantaneous (continuous) rate $\alpha$ and an expectation of $\alpha t$ changes per anatomical character over time $t$ [42]. Conversely, high rates of cladogenesis ($\lambda$) coupled with constant frequencies of punctuated (i.e. speciational) change [62] also would generate the patterns we documented. Under this model, we expect $\lambda t$ speciation events over time $t$. Given a frequency of punctuated anatomical change per speciation event $\varepsilon$, we would expect $\varepsilon\lambda t$ changes per anatomical character over time $t$ [63]. Thus, the continuous rate $\alpha$ used in this study comes very close to modelling $\varepsilon\lambda$, and we expect *estimated* $\alpha$ to decrease over time if $\lambda$ decreases over time even if $\varepsilon$ remains constant if the 'true' mode of change is punctuated rather than continuous. The punctuated model also allows for the fairly consistent species : genus ratios observed within the clade (electronic supplementary material, figure S6) and that the declining rates of anatomical change and cladogenesis should be similar. Although our best models suggest a nearly sevenfold decrease in $\lambda$ but only a three to four fold decrease in rates of anatomical change, we cannot reject declining $\lambda$ models with 'only' a twofold decrease in $\lambda$; we also cannot reject declining $\alpha$ models with as much as a fivefold decrease in $\alpha$ (or $\varepsilon\lambda$). Thus, our results are completely within the ranges expected given a constant rate of anatomical change per speciation event ($\varepsilon$)+

declining rates of cladogenesis ($\lambda$) over time model, which leads to two critical points. One, we can explain the overall pattern without any decline in the per-speciation rates of anatomical change. This is consistent with novel ecological opportunities, but not with low developmental constraints. Here, the high disparity is a side-effect of punctuated change coupled with elevated opportunities for speciation [61,64]. This stands in contrast to most early burst patterns in the fossil record, which better fit expectations of altered developmental pathways [65]. Two, interpretations of tip-dating analyses must consider both punctuated and continuous change models when testing hypotheses about the drivers of early bursts in disparity.

This paper represents one of the few examples of tip-dating methods being applied to taxa alive during the GOBE (e.g. [21]), and thus represents a first step in the application of this method to investigating the impact of environmental changes during this time period on evolutionary rates. Therefore, there are still many unanswered questions that are outside of the scope of this particular paper. It remains to be seen whether tip-dating analyses of clades that appear unaffected by the environmental shifts occurring from the Dapingian through the early Darriwillian (when analysed using taxonomic resampling methods) might show some shifts in rates of cladogenesis and/or anatomical evolution at this time. For example, while taxonomic resampling methods of the fossil record suggest that other brachiopod superfamilies such as the Triplesioidea or Camerelloidea [30] (see also electronic supplementary material, figure S13) did not experience a significant increase in diversity during this time period comparable to that of the Strophomenoidea, it is possible that tip-dating methods could reveal minor evolutionary rate shifts during the GOBE. Alternatively, such analyses might reveal that these clades responded to one or more of the other major environmental shifts that occurred during the Ordovician. This would be unsurprising if the GOBE was composed of a series of relatively closely occurring radiations caused by multiple environmental changes [2]. Furthermore, these rapid 'polyphyletic radiations' of many disparate and unrelated groups might be unique to the early Palaeozoic as there might not be subsequent examples of multiclade marine radiations except during rebounds from mass extinctions [66] (but see [8]). We require tip-dating analyses of marine taxa from throughout the Phanerozoic to assess this possibility. Clearly, robust evolutionary models that constrain not only the timing of diversification, but also rates of anatomical change and speciation simultaneously are key to understanding both the time scale and evolutionary mode of ecosystem response to environmental shifts.

**Data accessibility.** Code used in this project can be found on GitHub: https://github.com/PeterJWagner3/Supplementary/tree/main/Strophomenoids. R files, additional figures, and data used to conduct the analyses are available from the Dryad Digital Repository: https://doi.org/10.5061/dryad.k0p2ngf8g [67]. Discussion of our electronic supplementary material is available at Zenodo: https://zenodo.org/record/5033427#.YQrimUApBPY.

The data are provided in the electronic supplementary material [68].

**Authors' contributions.** C.R.C.: conceptualization, investigation, writing—original draft, writing—review and editing; M.E.P.: conceptualization, investigation, writing—original draft, writing—review and editing; P.J.W.: conceptualization, data curation, formal analysis, investigation, methodology, software, writing—original draft, writing—review and editing.

All authors gave final approval for publication and agreed to be held accountable for the work performed therein.

Competing interests. We declare we have no competing interests.

Funding. We received no funding for this study.

Acknowledgements. We thank B. Kröger, S. Holland, M. Foote, W. Kiessling, J. Alroy, A. Miller, S. Finnegan, A. Stigall, L. Ivany and P. Novack-Gottshall for entering relevant data into the Paleobiology Database. This is Paleobiology Database Publication 409. We also thank Christian Rasmussen and two anonymous reviewers for their comments and suggestions on a previous draft which greatly improved the quality of this paper.

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
