## [Peer Review File · Proceedings of the Royal Society B: Biological Sciences]

Review History

RSPB-2021-0092.R0 (Original submission)

Review form: Reviewer 1

Recommendation

Major revision is needed (please make suggestions in comments)

Scientific importance: Is the manuscript an original and important contribution to its field?

Good

General interest: Is the paper of sufficient general interest?

Good

Quality of the paper: Is the overall quality of the paper suitable?

Good

Is the length of the paper justified?

No

Should the paper be seen by a specialist statistical reviewer?

Yes

Do you have any concerns about statistical analyses in this paper? If so, please specify them explicitly in your report.

No

It is a condition of publication that authors make their supporting data, code and materials available - either as supplementary material or hosted in an external repository. Please rate, if applicable, the supporting data on the following criteria.

Is it accessible?

Yes

Is it clear?

Yes

Is it adequate?

Yes

Do you have any ethical concerns with this paper?

No

Comments to the Author

This is a paper with new idea, new methods and new results on the divergence times of Strophomenoidea during the GOBE. Strophomenoids had played an important role in the Ordovician brachiopod radiation, and its origination and early diversification had been investigated in great detail by several brachiopod experts since 1994. The authors are suggested to cite those works, and try to use fossil records to support the analytical results of this paper. Besides, several parts of this paper are not clearly and concisely stated that need to be modified.

Review form: Reviewer 2 (David Harper)

Recommendation

Accept with minor revision (please list in comments)

Scientific importance: Is the manuscript an original and important contribution to its field?

Excellent

General interest: Is the paper of sufficient general interest?

Excellent

Quality of the paper: Is the overall quality of the paper suitable?

Excellent

Is the length of the paper justified?

Yes

Should the paper be seen by a specialist statistical reviewer?

No

Do you have any concerns about statistical analyses in this paper? If so, please specify them explicitly in your report.

No

It is a condition of publication that authors make their supporting data, code and materials available - either as supplementary material or hosted in an external repository. Please rate, if applicable, the supporting data on the following criteria.

Is it accessible?

Yes

Is it clear?

Yes

Is it adequate?

Yes

Do you have any ethical concerns with this paper?

Yes

Comments to the Author

This is a landmark contribution applying the techniques of tip-dating and birth-death modelling to a well-known group of Palaeozoic brachiopods. The phylogenetic analysis, definition and coding of characters, is robust and associated graphical treatment clear and appropriate. My comments are minor.

There has been a tendency to focus on the Darriwilian interval as 'the GOBE' but other studies suggest the 'event' was more extended and more nuanced. Many contemporary studies are based on taxon counts and increasingly more sophisticated methods of data management. The finding that the radiation of this key group of strophomenides must be pushed back in time of course has implications for groups seen as precursors, for example the species-rich plectambonitoids (research in progress, Candela et al.), that may have their radiations pushed even further back in time too. And, of course there are further implications in the same way for understanding the stratigraphical relationships between the other key Ordovician brachiopod groups, the orthidine and dalmanellid brachiopods. The authors allude to the fact that despite searching for environmental triggers on a global scale, I guess invoking the Court Jester model, there may be a 'regionality' in clade radiations and at different time different branches may respond to more local factors, perhaps involving the Red Queen.

I have few comments but would first, pick up on the statement that the GOBE strongly affected only a minority of clades. Yes, if you use a narrow definition focused on only the Darriwilian; no, if you consider that significant radiations occurred during most of the period. Second, there are always difficulties in correlating changing environmental factors with diversifications (in back to back papers published some years ago in the Royal Society, diversifications were associated with global cooling and global warming). Commonly the significance of such correlations is based on speculation. In this case there is a 4 myr time lag between the diversification and increased oxygenation. The increased oxygenation may indeed have aided the diversification but may not have been the initial trigger. Perhaps adaptation to soft substrates preceded the Early Burst; it was already widespread across the plectambonitoids. Third, I have not tested this rigorously but the PBDB tends to favour sites in North America and Europe whereas the GBDB China. Can this be tested?

Review form: Reviewer 3 (Christian Rasmussen)

Recommendation

Accept with minor revision (please list in comments)

Scientific importance: Is the manuscript an original and important contribution to its field?

Excellent

General interest: Is the paper of sufficient general interest?

Good

Quality of the paper: Is the overall quality of the paper suitable?

Excellent

Is the length of the paper justified?

Yes

Should the paper be seen by a specialist statistical reviewer?

Yes

Do you have any concerns about statistical analyses in this paper? If so, please specify them explicitly in your report.

No

It is a condition of publication that authors make their supporting data, code and materials available - either as supplementary material or hosted in an external repository. Please rate, if applicable, the supporting data on the following criteria.

Is it accessible?

Yes

Is it clear?

Yes

Is it adequate?

Yes

Do you have any ethical concerns with this paper?

No

Comments to the Author

Review of the manuscript entitled "An Early Burst in brachiopod evolution corresponding with increasing oxygen-levels during the Great Ordovician Biodiversification Event" submitted to Royal Proceedings of the Royal Society B by Congreve et al.

This is a brilliant manuscript that uses novel techniques to constrain the origins and divergences times of one of the most important groups of Ordovician brachiopods, the strophomenida. Three different evolutionary models are applied and thoroughly tested, with one - the 'Early burst' model - being preferred by the authors based on their findings.

I find the results of this paper exiting and believe the methods applied are clearly explained. Novel findings include that the strophomenoid speciation bursts may have been previously understated by the literature and that the early burst model suggests the strophomenoids to originate in 3-4 myr before they appear in the rock record and basically place that split at the same point in time as Kröger et al (2019) showed the highest diversification rates for all metazoans during the Ordovician-Silurian interval (perhaps a reference that should be cited, by the way?). Both of these findings have great influence on how we perceive the GOBE-event - is it a slow, gradualistic rise in biodiversity accumulation (sensu Servais & Harper, 2018), or a punctualistic, rapid speciation phase (sensu Stigall et al., 2019). These new data certainly favour the latter hypothesis with advanced tip-dating techniques.

I therefore certainly think this manuscript is worthy of publication in this journal although I do have some remarks that I urge the authors to consider. I have no comments as to the techniques

applied. Or, actually, I do have one: it would be good if the authors could state or show from where geographically their data originate. It is based on the Paleobiology Database so I would think it is mainly Laurentian data (also as the earliest strophomenoids are from the Dapingian of China), but this is not explained anywhere.

I do, however, have considerable concerns regarding the authors 'correlation' to abiotic drivers. It is difficult for me to see how oxygen should create punctuated speciation. Arguably it would slowly open new niches, but why would brachiopods with low metabolic rates be dependent on that? Particularly if, as the authors state, strophomenoids were adapted to an infaunal mode.

Current knowledge on oxygenation of the sea during the Ordovician is quite uncertain. From the top of my head, only Tim Lenton and colleagues (PNAS, 2016) have looked at this but it seems the authors compare with atmospheric oxygen instead (Edwards et al. 2017) which is likely quite decoupled from oxygenation at the sea floor. I completely agree that we, of course, need to rely on the data at hand and the Edwards curve has a relatively high temporal resolution. In this case, though, I think there are three things the authors should consider before invoking oxygen as a causal mechanism:

1) If the earliest inclination point where atmospheric oxygen started to rise is in the earliest Darriwilian (as suggested by Edwards et al.), oxygen at the sea floor – where the strophomenoids lived infaunally in the sea bed – ought to have climbed much earlier than this point (i.e. way before the earliest Darriwilian). It would therefore precede your strophomenoid divergence time.

2) Rasmussen et al. 2019 (PNAS) went into depth with this apparent correlation between richness and oxygen (based on the Edwards curve) during the earliest Darriwilian and showed, by use of first differences, that an earliest Darriwilian rise across all metazoans seemed to better correlate with the temperature curve of Trotter et al. (2008, Science). As I read your manuscript, your divergence times fall exactly within the same Dw1-time slice as Rasmussen et al. found the rise in richness to occur in and to be best coupled to cooling climate. Also, this is where SST's reach modern values which was suggested by Rasmussen et al 2019 to be the primary causal mechanism behind the radiations. Your data is in good agreement with that, as I see it. See also attached figure where I try to illustrate this point by overlaying the Rasmussen et al fig on top of your divergence times. It seems you might be using the Grossmann (2020)-curve temperature curve although you refer (indirectly) to Trotter et al., 2008? If you use the Trotter curve (which I would suggest you should) please pay attention to the early Darriwilian low in that curve – it seems missing in your figure.

3) Even if the rise in oxygen did coincide with the main strophomenoid divergence, that rise would likely be slow (as mentioned above). This does not correlate with a burst mode of speciation.

Based on the above, I think it is much more likely that climate was the driver of your burst mode speciation simply through controlling pathways in intra-cratonic basins (see, for instance, Stigall et al., 2017; Pedersen & Rasmussen 2019). To me your early burst model divergence times match exactly the sea level curve you inserted which supposedly should be from Rasmussen et al 2019. If so, it is a 3rd order sea level curve, again implying climate as your main driver, not oxygen.

Even if we were to take the fossil record at face value where the origin of the strophomenoids may come in a bit late (late Dw2) this too seems to correlate with the later Darriwilian cooling pulse near the MDICE-interval (Fang et al., 2019, Global & Planetary Change).

In summary, I urge the authors to reassess their correlation to abiotic drivers and thus reconsider if their title is appropriate – in my view it should not highlight oxygen, but rather climate (or, perhaps just 'abiotic drivers' if the authors are not agreeing with the above) – as well as the manuscript text accordingly (particularly chapter 4).

Minor edits:

- l. 69 and elsewhere: duration (in this case 'figures 1-2') is usually denoted by an en dash (-), not hyphens (-)
- l. 158: the early Palaeozoic is not a formalized unit. Thus, it should be written in lower case. I.e. 'early', not 'Early'.
- l. 210: this is an extremely exciting finding and one that potentially could have implications for everything: what would it look like for other clades, for instance? This timing is highly likely, though, given the earliest strophomenoids are known from the Dapingian in China (Zhan et al., 2013, *Palaeontology*).
- l. 244: here the authors frame the paradox; why should a rapid diversification be best explained by a 'long-term trend towards increased oxygen'? To me this is better explained by climatic oscillations on the kyr-scale that affect sea level, thus favouring allopatric speciation rather than a slow, steady rise in oxygen on the myr-scale. Chapter 4 Should therefore be modified in my view.
- l. 250: Perhaps of relevance here Rasmussen et al., 2016 suggested the MDICE to be a result of exactly that - increased bioproductivity.
- l. 250: The reference to Bergström et al. 2009 seems out of place here. Do you perhaps mean ref. 46?
- l. 256-258: Is it intentional to shift the font to italics?
- l. 341: Not sure what you mean by this sentence? The reference you refer to (Harper et al., 2013), show that many clades were greatly affected by the GOBE. Notably the strophomenida. I therefore do not agree that the implications of your findings are 'limited'. I think this is a really, really important contribution as we do not have much other data showing phylogenies with divergence times (apart from perhaps from graptolites) that show similar level of temporal detail. Captions, fig S7: I believe the strophomenoid stratigraphical ranges are orange, not yellow?

See also attached figure. I would be happy to share my illustrator-file of that figure should the authors be interested in precisely redrawing the abiotic drivers using the original file next to the divergence figure.

-Christian Rasmussen

Decision letter (RSPB-2021-0092.R0)

02-Mar-2021

Dear Dr Congreve:

I am writing to inform you that your manuscript RSPB-2021-0092 entitled "An Early Burst in brachiopod evolution corresponding with increasing oxygen-levels during the Great Ordovician Biodiversification Event" has, in its current form, been rejected for publication in *Proceedings B*.

This action has been taken on the advice of referees, who have recommended that substantial revisions are necessary. With this in mind we would be happy to consider a resubmission, provided the comments of the referees are fully addressed. However please note that this is not a provisional acceptance.

Please note that this decision may (or may not) have taken into account confidential comments.

In your revision process, please take a second look at how open your science is; our policy is that *ALL* (maximally inclusive) data involved with the study should be made openly accessible, fully enabling re-use, replication and transparency-- see:

<https://royalsociety.org/journals/ethics-policies/data-sharing-mining/>

Insufficient sharing of data can delay or even cause rejection of a paper.

Full data and code/scripts to enable reuse/replication/repurposing are what this policy intends.

Sincerely,

Dr John Hutchinson, Editor

Associate Editor

Board Member: 1

Comments to Author:

Thank you for the opportunity to read this study. Herein the authors examine the correlation between the evolution of a major sub-clade of brachiopods and global environmental change over geological timescales. They recover interesting patterns in terms of speciation and anatomical disparity over time in their study group, which in itself provides important insight in brachiopod evolutionary dynamics. However, they also correlate these dynamics with changes in global oxygen and sea levels. This is therefore a very interesting and impactful analysis that is likely to be of broad interest to evolutionary biologists, palaeontologists and earth system scientists. I'm not an expert in the group under study here, but the expert reviewers are on the whole complementary about the study, the methods employed and the results. However, the reviewers do suggest a number of changes that would improve the manuscript. Reviewer 1's comments are unfortunately a little vague and it might therefore be challenge for the authors to respond, beyond doing their best to ensure that the wider literature is cited as appropriately as possible.

Reviewer(s)' Comments to Author:

Referee: 1

Comments to the Author(s)

This is a paper with new idea, new methods and new results on the divergence times of Strophomenoidea during the GOBE. Strophomenoids had played an important role in the Ordovician brachiopod radiation, and its origination and early diversification had been investigated in great detail by several brachiopod experts since 1994. The authors are suggested

to cite those works, and try to use fossil records to support the analytical results of this paper. Besides, several parts of this paper are not clearly and concisely stated that need to be modified.

Referee: 2

Comments to the Author(s)

This is a landmark contribution applying the techniques of tip-dating and birth-death modelling to a well-known group of Palaeozoic brachiopods. The phylogenetic analysis, definition and coding of characters, is robust and associated graphical treatment clear and appropriate. My comments are minor.

There has been a tendency to focus on the Darriwilian interval as 'the GOBE' but other studies suggest the 'event' was more extended and more nuanced. Many contemporary studies are based on taxon counts and increasingly more sophisticated methods of data management. The finding that the radiation of this key group of strophomenides must be pushed back in time of course has implications for groups seen as precursors, for example the species-rich plectambonitoids (research in progress, Candela et al.), that may have their radiations pushed even further back in time too. And, of course there are further implications in the same way for understanding the stratigraphical relationships between the other key Ordovician brachiopod groups, the orthidine and dalmanellidine brachiopods. The authors allude to the fact that despite searching for environmental triggers on a global scale, I guess invoking the Court Jester model, there may be a 'regionality' in clade radiations and at different time different branches may respond to more local factors, perhaps involving the Red Queen.

I have few comments but would first, pick up on the statement that the GOBE strongly affected only a minority of clades. Yes, if you use a narrow definition focused on only the Darriwilian; no, if you consider that significant radiations occurred during most of the period. Second, there are always difficulties in correlating changing environmental factors with diversifications (in back to back papers published some years ago in the Royal Society, diversifications were associated with global cooling and global warming). Commonly the significance of such correlations is based on speculation. In this case there is a 4 myr time lag between the diversification and increased oxygenation. The increased oxygenation may indeed have aided the diversification but may not have been the initial trigger. Perhaps adaptation to soft substrates preceded the Early Burst; it was already widespread across the plectambonitoids. Third, I have not tested this rigorously but the PBDB tends to favour sites in North America and Europe whereas the GBDB China. Can this be tested?

Referee: 3

Comments to the Author(s)

Review of the manuscript entitled "An Early Burst in brachiopod evolution corresponding with increasing oxygen-levels during the Great Ordovician Biodiversification Event" submitted to Royal Proceedings of the Royal Society B by Congreve et al.

This is a brilliant manuscript that uses novel techniques to constrain the origins and divergences times of one of the most important groups of Ordovician brachiopods, the strophomenida. Three different evolutionary models are applied and thoroughly tested, with one - the 'Early burst' model - being preferred by the authors based on their findings.

I find the results of this paper exiting and believe the methods applied are clearly explained. Novel findings include that the strophomenoid speciation bursts may have been previously understated by the literature and that the early burst model suggests the strophomenoids to originate in 3-4 myr before they appear in the rock record and basically place that split at the same point in time as Kröger et al (2019) showed the highest diversification rates for all metazoans during the Ordovician-Silurian interval (perhaps a reference that should be cited, by the way?). Both of these findings have great influence on how we perceive the GOBE-event - is it a slow, gradualistic rise in biodiversity accumulation (sensu Servais & Harper, 2018), or a punctualistic, rapid speciation phase (sensu Stigall et al., 2019). These new data certainly favour the latter hypothesis with advanced tip-dating techniques.

I therefore certainly think this manuscript is worthy of publication in this journal although I do have some remarks that I urge the authors to consider. I have no comments as to the techniques applied. Or, actually, I do have one: it would be good if the authors could state or show from where geographically their data originate. It is based on the Paleobiology Database so I would think it is mainly Laurentian data (also as the earliest strophomenoids are from the Dapingian of China), but this is not explained anywhere.

I do, however, have considerable concerns regarding the authors 'correlation' to abiotic drivers. It is difficult for me to see how oxygen should create punctuated speciation. Arguably it would slowly open new niches, but why would brachiopods with low metabolic rates be dependent on that? Particularly if, as the authors state, strophomenoids were adapted to an infaunal mode.

Current knowledge on oxygenation of the sea during the Ordovician is quite uncertain. From the top of my head, only Tim Lenton and colleagues (PNAS, 2016) have looked at this but it seems the authors compare with atmospheric oxygen instead (Edwards et al. 2017) which is likely quite decoupled from oxygenation at the sea floor. I completely agree that we, of course, need to rely on the data at hand and the Edwards curve has a relatively high temporal resolution. In this case, though, I think there are three things the authors should consider before invoking oxygen as a causal mechanism:

1) If the earliest inclination point where atmospheric oxygen started to rise is in the earliest Darriwilian (as suggested by Edwards et al.), oxygen at the sea floor – where the strophomenoids lived infaunally in the sea bed – ought to have climbed much earlier than this point (i.e. way before the earliest Darriwilian). It would therefore precede your strophomenoid divergence time.

2) Rasmussen et al. 2019 (PNAS) went into depth with this apparent correlation between richness and oxygen (based on the Edwards curve) during the earliest Darriwilian and showed, by use of first differences, that an earliest Darriwilian rise across all metazoans seemed to better correlate with the temperature curve of Trotter et al. (2008, Science). As I read your manuscript, your divergence times fall exactly within the same Dw1-time slice as Rasmussen et al. found the rise in richness to occur in and to be best coupled to cooling climate. Also, this is where SST's reach modern values which was suggested by Rasmussen et al 2019 to be the primary causal mechanism behind the radiations. Your data is in good agreement with that, as I see it. See also attached figure where I try to illustrate this point by overlaying the Rasmussen et al fig on top of your divergence times. It seems you might be using the Grossmann (2020)-curve temperature curve although you refer (indirectly) to Trotter et al., 2008? If you use the Trotter curve (which I would suggest you should) please pay attention to the early Darriwilian low in that curve – it seems missing in your figure.

3) Even if the rise in oxygen did coincide with the main strophomenoid divergence, that rise would likely be slow (as mentioned above). This does not correlate with a burst mode of speciation.

Based on the above, I think it is much more likely that climate was the driver of your burst mode speciation simply through controlling pathways in intra-cratonic basins (see, for instance, Stigall et al., 2017; Pedersen & Rasmussen 2019). To me your early burst model divergence times match exactly the sea level curve you inserted which supposedly should be from Rasmussen et al 2019. If so, it is a 3rd order sea level curve, again implying climate as your main driver, not oxygen.

Even if we were to take the fossil record at face value where the origin of the strophomenoids may come in a bit late (late Dw2) this too seems to correlate with the later Darriwilian cooling pulse near the MDICE-interval (Fang et al., 2019, Global & Planetary Change).

In summary, I urge the authors to reassess their correlation to abiotic drivers and thus reconsider if their title is appropriate – in my view it should not highlight oxygen, but rather climate (or,

perhaps just 'abiotic drivers' if the authors are not agreeing with the above) – as well as the manuscript text accordingly (particularly chapter 4).

Minor edits:

l. 69 and elsewhere: duration (in this case 'figures 1–2') is usually denoted by an en dash (–), not hyphens (-)

l. 158: the early Palaeozoic is not a formalized unit. Thus, it should be written in lower case. I.e. 'early', not 'Early'.

l. 210: this is an extremely exciting finding and one that potentially could have implications for everything: what would it look like for other clades, for instance? This timing is highly likely, though, given the earliest strophomenoids are known from the Dapingian in China (Zhan et al., 2013, Palaeontology).

l. 244: here the authors frame the paradox; why should a rapid diversification be best explained by a 'long-term trend towards increased oxygen'? To me this is better explained by climatic oscillations on the kyr-scale that affect sea level, thus favouring allopatric speciation rather than a slow, steady rise in oxygen on the myr-scale. Chapter 4 Should therefore be modified in my view.

l. 250: Perhaps of relevance here Rasmussen et al., 2016 suggested the MDICE to be a result of exactly that – increased bioproductivity.

l. 250: The reference to Bergström et al. 2009 seems out of place here. Do you perhaps mean ref. 46?

l. 256–258: Is it intentional to shift the font to italics?

l. 341: Not sure what you mean by this sentence? The reference you refer to (Harper et al., 2013), show that many clades were greatly affected by the GOBE. Notably the strophomenida. I therefore do not agree that the implications of your findings are 'limited'. I think this is a really, really important contribution as we do not have much other data showing phylogenies with divergence times (apart from perhaps from graptolites) that show similar level of temporal detail. Captions, fig S7: I believe the strophomenoid stratigraphical ranges are orange, not yellow?

See also attached figure. I would be happy to share my illustrator-file of that figure should the authors be interested in precisely redrawing the abiotic drivers using the original file next to the divergence figure.

-Christian Rasmussen

Author's Response to Decision Letter for (RSPB-2021-0092.R0)

See Appendix A.

RSPB-2021-1450.R0

Review form: Reviewer 1

Recommendation

Major revision is needed (please make suggestions in comments)

Scientific importance: Is the manuscript an original and important contribution to its field?

Acceptable

General interest: Is the paper of sufficient general interest?

Acceptable

Quality of the paper: Is the overall quality of the paper suitable?

Acceptable

Is the length of the paper justified?

No

Should the paper be seen by a specialist statistical reviewer?

Yes

Do you have any concerns about statistical analyses in this paper? If so, please specify them explicitly in your report.

Yes

It is a condition of publication that authors make their supporting data, code and materials available - either as supplementary material or hosted in an external repository. Please rate, if applicable, the supporting data on the following criteria.

Is it accessible?

Yes

Is it clear?

Yes

Is it adequate?

No

Do you have any ethical concerns with this paper?

No

Comments to the Author

The origin and early diversification of Strophomenoidea is very important to the early diversification of Brachiopoda and to the GOBE. This manuscript makes an important contribution to these topics, particularly to the further study of GOBE, because the authors are using some new methods such as Bayesian Tip-Dating method. But, unfortunately, there are several important relevant references that are missed by the authors, which makes the "Discussion and Conclusion" not reliable! Besides, the expression of the entire manuscript could be more concise and easier to be understood.

Review form: Reviewer 3 (Christian Rasmussen)

Recommendation

Accept as is

Scientific importance: Is the manuscript an original and important contribution to its field?

Excellent

General interest: Is the paper of sufficient general interest?

Excellent

Quality of the paper: Is the overall quality of the paper suitable?

Excellent

Is the length of the paper justified?

Yes

Should the paper be seen by a specialist statistical reviewer?

No

Do you have any concerns about statistical analyses in this paper? If so, please specify them explicitly in your report.

No

It is a condition of publication that authors make their supporting data, code and materials available - either as supplementary material or hosted in an external repository. Please rate, if applicable, the supporting data on the following criteria.

Is it accessible?

Yes

Is it clear?

Yes

Is it adequate?

Yes

Do you have any ethical concerns with this paper?

No

Comments to the Author

This revised version of the manuscript address all concerns raised by me and I, therefore, have no hesitations in recommending this fine manuscript for publication.

I have a few minor edits, though:

l. 33: the 'Great Ordovician Biodiversification Event'-keyword is already used in the title and should thus not be a keyword. Perhaps change to the acronym 'GOBE'

l. 42: '3-5' (use an en dash instead of hyphen)

l. 49: write out 'Paleozoic Evolutionary Fauna' and use US-English as this term was first defined by Sepkoski in Paleobiology

l. 49: '10-14' (use en dash instead of hyphen)

l. 50: '15-21' (use en dash instead of hyphen)

l. 61: '~470-458' (use en dash instead of hyphen)

l. 62: There is a double space after the full stop

l. 67: '1-2' (use en dash instead of hyphen)

l. 134: '9-11' (use en dash instead of hyphen)

l. 170: '4-6' (use en dash instead of hyphen)

l. 216: Given the importance of this species for your analysis I suggest also citing Zhan et al., 2013 (Palaeontology, Vol. 56, Part 5, 2013, pp. 1121-1148)

l. 217: '7-8' (use en dash instead of hyphen)

l. 231: '3-3.5' (use en dash instead of hyphen)

l. 249: the delta sign is corrupted in my version, so please check if it looks correct

l. 382: 'Darriwilian', not 'Dariwillian'

l. 361: '3-4' (use en dash instead of hyphen)

l. 391: there seems to be a double space after the full stop.

l. 413: 'Rasmussen', not 'Rassmussen'

Figures:

Looks great.

References in general:

References have not been checked in detail but there is no consistency as to the use of hyphens and en dash when citing page numbers. Use en dash consistently. Also, in some cases, spaces are included on either side of the hyphen/ en dash whereas in other cases there are no spaces added.

Lastly, I would like to congratulate the authors - this is a great study and I look forward to seeing it published!

-Christian Rasmussen

Decision letter (RSPB-2021-1450.R0)

27-Jul-2021

Dear Dr Congreve

I am pleased to inform you that your manuscript RSPB-2021-1450 entitled "An Early Burst in brachiopod evolution corresponding with significant climatic shifts during the Great Ordovician Biodiversification Event" has been accepted for publication in Proceedings B. Congratulations!!

The referee(s) have recommended publication, but also suggest some minor revisions to your manuscript. Therefore, I invite you to respond to the referee(s)' comments and revise your manuscript. Because the schedule for publication is very tight, it is a condition of publication that you submit the revised version of your manuscript within 7 days. If you do not think you will be able to meet this date please let us know.

As the Associate Editor notes, one reviewer urged some unspecified changes but the other made some more specific suggestions on the prose, and we think you should be able to satisfy both of them with some further referencing and honing of the writing style.

- 1) A text file of the manuscript (doc, txt, rtf or tex), including the references, tables (including captions) and figure captions. Please remove any tracked changes from the text before submission. PDF files are not an accepted format for the "Main Document".
- 2) A separate electronic file of each figure (tiff, EPS or print-quality PDF preferred). The format should be produced directly from original creation package, or original software format. PowerPoint files are not accepted.

3) Electronic supplementary material: this should be contained in a separate file and where possible, all ESM should be combined into a single file. All supplementary materials accompanying an accepted article will be treated as in their final form. They will be published alongside the paper on the journal website and posted on the online figshare repository. Files on figshare will be made available approximately one week before the accompanying article so that the supplementary material can be attributed a unique DOI.

Sincerely,

Dr John Hutchinson

Associate Editor

Board Member

Comments to Author:

I agree with the reviewers that, firstly, the original round of reviews have largely been addressed, and secondly that this paper makes a significant contribution to the literature. Reviewer 2 provides a list of minor typographical changes that would improve a revised version. Reviewer 1 again asks if the discussion could be more concisely presented, and for wider citation of the literature but does not provide any specific guidance in either case. I would suggest that authors look at the reference list in the paper reviewer 1 cites and use google scholar to examine papers that been published since that cite this paper, and subsequently add any relevant studies where appropriate.

Reviewer(s)' Comments to Author:

Referee: 1

Comments to the Author(s).

The origin and early diversification of Strophomenoidea is very important to the early diversification of Brachiopoda and to the GOBE. This manuscript makes an important contribution to these topics, particularly to the further study of GOBE, because the authors are using some new methods such as Bayesian Tip-Dating method. But, unfortunately, there are several important relevant references that are missed by the authors, which makes the "Discussion and Conclusion" not reliable! Besides, the expression of the entire manuscript could be more concise and easier to be understood.

Referee: 3

Comments to the Author(s).

This revised version of the manuscript address all concerns raised by me and I, therefore, have no hesitations in recommending this fine manuscript for publication.

I have a few minor edits, though:

l. 33: the 'Great Ordovician Biodiversification Event'-keyword is already used in the title and should thus not be a keyword. Perhaps change to the acronym 'GOBE'

l. 42: '3-5' (use an en dash instead of hyphen)

l. 49: write out 'Paleozoic Evolutionary Fauna' and use US-English as this term was first defined by Sepkoski in Paleobiology

l. 49: '10-14' (use en dash instead of hyphen)

l. 50: '15-21' (use en dash instead of hyphen)

l. 61: '~470-458' (use en dash instead of hyphen)

l. 62: There is a double space after the full stop

l. 67: '1-2' (use en dash instead of hyphen)

l. 134: '9-11' (use en dash instead of hyphen)

l. 170: '4-6' (use en dash instead of hyphen)

l. 216: Given the importance of this species for your analysis I suggest also citing Zhan et al., 2013 (Palaeontology, Vol. 56, Part 5, 2013, pp. 1121-1148)

l. 217: '7-8' (use en dash instead of hyphen)

l. 231: '3-3.5' (use en dash instead of hyphen)

l. 249: the delta sign is corrupted in my version, so please check if it looks correct

l. 382: 'Darrivilian', not 'Dariwillian'

l. 361: '3-4' (use en dash instead of hyphen)

l. 391: there seems to be a double space after the full stop.

l. 413: 'Rasmussen', not 'Rassmussen'

Figures:

Looks great.

References in general:

References have not been checked in detail but there is no consistency as to the use of hyphens and en dash when citing page numbers. Use en dash consistently. Also, in some cases, spaces are included on either side of the hyphen/ en dash whereas in other cases there are no spaces added.

Lastly, I would like to congratulate the authors - this is a great study and I look forward to seeing it published!

-Christian Rasmussen

Author's Response to Decision Letter for (RSPB-2021-1450.R0)

See Appendix B.

Decision letter (RSPB-2021-1450.R1)

06-Aug-2021

Dear Dr Congreve

I am pleased to inform you that your manuscript entitled "An Early Burst in brachiopod evolution corresponding with significant climatic shifts during the Great Ordovician Biodiversification Event" has been accepted for publication in Proceedings B.

Your article has been estimated as being 9 pages long. Our Production Office will be able to confirm the exact length at proof stage.

Data Accessibility section

Open Access

Paper charges

Sincerely,
Editor, Proceedings B
mailto: proceedingsb@royalsociety.org

Appendix A

To the editors of the Proceedings of the Royal Society B,

We would like to resubmit our manuscript, “An Early Burst in brachiopod evolution corresponding with significant climatic shifts during the Great Ordovician Biodiversification Event,” for consideration as a research article. This paper uses fossilized birth-death modeling to estimate the rates of cladogenesis and the timing of diversification of strophomenoid brachiopods during the Great Ordovician Biodiversification Event, one of the largest evolutionary radiations in Earth’s history. We have addressed the comments from the three referees in our resubmission. Below we have included a copy of the original reviews, along with our replies to each referee’s comments in bold text.

Thank you very much for your consideration.

Sincerely,

Curtis R Congreve, Peter J. Wagner, and Mark E. Patzkowsky

Associate Editor

Board Member: 1

Comments to Author:

Thank you for the opportunity to read this study. Herein the authors examine the correlation between the evolution of a major sub-clade of brachiopods and global environmental change over geological timescales. They recover interesting patterns in terms of speciation and anatomical disparity over time in their study group, which in itself provides important insight in brachiopod evolutionary dynamics. However, they also correlate these dynamics with changes in global oxygen and sea levels. This is therefore a very interesting and impactful analysis that is likely to be of broad interest to evolutionary biologists, palaeontologists and earth system scientists. I’m not an expert in the group under study here, but the expert reviewers are on the whole complementary about the study, the methods employed and the results. However, the reviewers do suggest a number of changes that would improve the manuscript. Reviewer 1’s comments are unfortunately a little vague and it might therefore be challenge for the authors to respond, beyond doing their best to ensure that the wider literature is cited as appropriately as possible.

Reviewer(s)' Comments to Author:

Referee: 1

Comments to the Author(s)

This is a paper with new idea, new methods and new results on the divergence times of Strophomenoidea during the GOBE. Strophomenoids had played an important role in the Ordovician brachiopod radiation, and its origination and early diversification had been investigated in great detail by several brachiopod experts since 1994. The authors are suggested to cite those works, and try to use fossil records to support the analytical results of

this paper. Besides, several parts of this paper are not clearly and concisely stated that need to be modified.

We have made significant revisions to the discussion (as requested by Reviewers 2 and 3) which have added clarity to this part of the manuscript. We have also added some additional references on the taxonomic history of the strophomenoids, particularly the mentioned paper Rong and Cocks 1994. As far as including the fossil record, our analyses have worked with the most up to date occurrence information and taxonomic genus level assignments from the PBDB, and we have also added additional information to the PBDB to fill in missing information. This has been clarified in the description of our methods.

Referee: 2

Comments to the Author(s)

This is a landmark contribution applying the techniques of tip-dating and birth-death modelling to a well-known group of Palaeozoic brachiopods. The phylogenetic analysis, definition and coding of characters, is robust and associated graphical treatment clear and appropriate. My comments are minor.

There has been a tendency to focus on the Darriwilian interval as 'the GOBE' but other studies suggest the 'event' was more extended and more nuanced. Many contemporary studies are based on taxon counts and increasingly more sophisticated methods of data management. The finding that the radiation of this key group of strophomenides must be pushed back in time of course has implications for groups seen as precursors, for example the species-rich plectambonitoids (research in progress, Candela et al.), that may have their radiations pushed even further back in time too. And, of course there are further implications in the same way for understanding the stratigraphical relationships between the other key Ordovician brachiopod groups, the orthidine and dalmanellidine brachiopods. The authors allude to the fact that despite searching for environmental triggers on a global scale, I guess invoking the Court Jester model, there may be a 'regionality' in clade radiations and at different time different branches may respond to more local factors, perhaps involving the Red Queen.

I have few comments but would first, pick up on the statement that the GOBE strongly affected only a minority of clades. Yes, if you use a narrow definition focused on only the Darriwilian; no, if you consider that significant radiations occurred during most of the period. Second, there are always difficulties in correlating changing environmental factors with diversifications (in back to back papers published some years ago in the Royal Society, diversifications were associated with global cooling and global warming). Commonly the significance of such correlations is based on speculation. In this case there is a 4 myr time lag between the diversification and increased oxygenation. The increased oxygenation may indeed have aided the diversification but may not have been the initial trigger. Perhaps adaptation to soft substrates preceded the Early Burst; it was already widespread across the plectambonitoids. Third, I have not tested this rigorously but the PBDB tends to favour sites in North America and Europe whereas the GBDB China. Can this be tested?

We agree that geographic biases are an incredibly important point. We attempted to fill known gaps in our record by adding additional material to the PBDB (which has been stated in the text). However, an analysis of geographic variation in rates changes is something that is outside of the scope of this particular paper (though it is a project we are interested in pursuing in an additional paper).

We have added a brief discussion of the Plectambonitoids in the text as an additional consideration. The increased evolutionary rates is seen in most of the Strophomenida (including the Plectams) which does indeed have potential impact on our interpretations of the timing of evolutionary radiations further down the tree. Our discussion here is minor because, similar to the issue above, it is a fascinating question that is unfortunately outside of the scope of this paper.

Our analysis did not return a timing of increased diversification that fit strongly with many of the proposed biologically mediated mechanisms for radiation; either the increases in productivity which could be the cause of the carbon isotopic shift in the late Dapingian or the development of reef ecosystems to better facilitate the expansion of soft sediment environments. While this does not rule out the possibility of biotically important mechanisms of increased evolutionary change, we have chosen to focus on the environmental stimuli given that the other biological changes which have been previously suggested as being important are occurring well after the period of time currently designated by Rasmussen and others as the GOBE.

Referee: 3

Comments to the Author(s)

Review of the manuscript entitled "An Early Burst in brachiopod evolution corresponding with increasing oxygen-levels during the Great Ordovician Biodiversification Event" submitted to Royal Proceedings of the Royal Society B by Congreve et al.

This is a brilliant manuscript that uses novel techniques to constrain the origins and divergences times of one of the most important groups of Ordovician brachiopods, the strophomenida. Three different evolutionary models are applied and thoroughly tested, with one – the 'Early burst' model – being preferred by the authors based on their findings.

I find the results of this paper exiting and believe the methods applied are clearly explained. Novel findings include that the strophomenoid speciation bursts may have been previously understated by the literature and that the early burst model suggests the strophomenoids to originate in 3–4 myr before they appear in the rock record and basically place that split at the same point in time as Kröger et al (2019) showed the highest diversification rates for all metazoans during the Ordovician–Silurian interval (perhaps a reference that should be cited, by the way?). Both of these findings have great influence on how we perceive the GOBE-event – is it a slow, gradualistic rise in biodiversity accumulation (sensu Servais & Harper, 2018), or a punctualistic, rapid speciation phase (sensu Stigall et al., 2019). These new data certainly favour the latter hypothesis with advanced tip-dating techniques.

I therefore certainly think this manuscript is worthy of publication in this journal although I do have some remarks that I urge the authors to consider. I have no comments as to the techniques applied. Or, actually, I do have one: it would be good if the authors could state or show from where geographically their data originate. It is based on the Paleobiology Database so I would think it is mainly Laurentian data (also as the earliest strophomenoids are from the Dapingian of China), but this is not explained anywhere.

The PBDB database is a global database with sampling from around the world. At this time period there is significant sampling not just in Laurentia, but also Baltica and China. We have also added additional occurrence data to the database whenever we have seen

areas that were significantly lacking in representation. We have amended our supplemental information to include sampling information from different geographic provinces.

I do, however, have considerable concerns regarding the authors 'correlation' to abiotic drivers. It is difficult for me to see how oxygen should create punctuated speciation. Arguably it would slowly open new niches, but why would brachiopods with low metabolic rates be dependent on that? Particularly if, as the authors state, strophomenids were adapted to an infaunal mode.

We have revised our discussion on causal mechanisms to better reflect the diversity of different environmental triggers for this extinction. We have taken a more agnostic approach, especially since pluralistic interactions between different environmental changes could be necessary to facilitate increased speciation observed during this time period.

Current knowledge on oxygenation of the sea during the Ordovician is quite uncertain. From the top of my head, only Tim Lenton and colleagues (PNAS, 2016) have looked at this but it seems the authors compare with atmospheric oxygen instead (Edwards et al. 2017) which is likely quite decoupled from oxygenation at the sea floor. I completely agree that we, of course, need to rely on the data at hand and the Edwards curve has a relatively high temporal resolution. In this case, though, I think there are three things the authors should consider before invoking oxygen as a causal mechanism:

1) If the earliest inclination point where atmospheric oxygen started to rise is in the earliest Darriwilian (as suggested by Edwards et al.), oxygen at the sea floor – where the strophomenoids lived infaunally in the sea bed – ought to have climbed much earlier than this point (i.e. way before the earliest Darriwilian). It would therefore precede your strophomenoid divergence time.

2) Rasmussen et al. 2019 (PNAS) went into depth with this apparent correlation between richness and oxygen (based on the Edwards curve) during the earliest Darriwilian and showed, by use of first differences, that an earliest Darriwilian rise across all metazoans seemed to better correlate with the temperature curve of Trotter et al. (2008, Science). As I read your manuscript, your divergence times fall exactly within the same Dw1-time slice as Rasmussen et al. found the rise in richness to occur in and to be best coupled to cooling climate. Also, this is where SST's reach modern values which was suggested by Rasmussen et al 2019 to be the primary causal mechanism behind the radiations. Your data is in good agreement with that, as I see it. See also attached figure where I try to illustrate this point by overlaying the Rasmussen et al fig on top of your divergence times. It seems you might be using the Grossmann (2020)-curve temperature curve although you refer (indirectly) to Trotter et al., 2008? If you use the Trotter curve (which I would suggest you should) please pay attention to the early Darriwilian low in that curve – it seems missing in your figure.

We have amended our figure based on the recommendation of Rasmussen. With the new figure, we have also updated our discussion on environmental triggers, taking more time to discuss the importance of sea level rise and fall, as well as the importance of global cooling to current sea surface temperature ranges.

3) Even if the rise in oxygen did coincide with the main strophomenoid divergence, that rise would likely be slow (as mentioned above). This does not correlate with a burst mode of speciation.

Based on the above, I think it is much more likely that climate was the driver of your burst mode speciation simply through controlling pathways in intra-cratonic basins (see, for instance, Stigall et al., 2017; Pedersen & Rasmussen 2019). To me your early burst model divergence times match exactly the sea level curve you inserted which supposedly should be from Rasmussen et al 2019. If so, it is a 3rd order sea level curve, again implying climate as your main driver, not oxygen.

Again, as mentioned earlier, our discussion on the environmental triggers has been altered to allow for a more pluralistic discussion of environmental triggers, as well as discussion that the “cause” of this shift may be synergistic reactions between multiple environmental changes which in aggregate facilitate higher rates of speciation (as suggested in Stigall et al 2019)

Even if we were to take the fossil record at face value where the origin of the strophomenoids may come in a bit late (late Dw2) this too seems to correlate with the later Darriwilian cooling pulse near the MDICE-interval (Fang et al., 2019, Global & Planetary Change).

In summary, I urge the authors to reassess their correlation to abiotic drivers and thus reconsider if their title is appropriate – in my view it should not highlight oxygen, but rather climate (or, perhaps just ‘abiotic drivers’ if the authors are not agreeing with the above) – as well as the manuscript text accordingly (particularly chapter 4).

As mentioned both above and below, we have significantly revised the discussion section to address the importance of multiple environmental shifts during this time period and not simply highlight oxygen at the expense of others.

Minor edits:

l. 69 and elsewhere: duration (in this case ‘figures 1–2’) is usually denoted by an en dash (–), not hyphens (-)

l. 158: the early Palaeozoic is not a formalized unit. Thus, it should be written in lower case. I.e. ‘early’, not ‘Early’.

Fixed

l. 210: this is an extremely exiting finding and one that potentially could have implications for everything: what would it look like for other clades, for instance? This timing is highly likely, though, given the earliest strophomenoids are known from the Dapingian in China (Zhan et al., 2013, Palaeontology).

l. 244: here the authors frame the paradox; why should a rapid diversification be best explained by a ‘long-term trend towards increased oxygen’? To me this is better explained by climatic oscillations on the kyr-scale that affect sea level, thus favouring allopatric speciation rather than a slow, steady rise in oxygen on the myr-scale. Chapter 4 Should therefore be modified in my view.

As mentioned above, this section has been significantly revised.

l. 250: Perhaps of relevance here Rasmussen et al., 2016 suggested the MDICE to be a result of exactly that – increased bioproductivity.

As mentioned above, much of this section has been completely revised. On the subject of productivity, we have made a note to acknowledge increased productivity as a potential cause of the changes in carbon isotopic composition through the Ordovician.

l. 250: The reference to Bergström et al. 2009 seems out of place here. Do you perhaps mean ref. 46?

The has been corrected and in fact this entire section has been substantially revised

l. 256–258: Is it intentional to shift the font to italics?

This section has been highly revised and this section is no longer in italics.

l. 341: Not sure what you mean by this sentence? The reference you refer to (Harper et al., 2013), show that many clades were greatly affected by the GOBE. Notably the strophomenida. I therefore do not agree that the implications of your findings are 'limited'. I think this is a really, really important contribution as we do not have much other data showing phylogenies with divergence times (apart from perhaps from graptolites) that show similar level of temporal detail. Captions, fig S7: I believe the strophomenoid stratigraphical ranges are orange, not yellow?

We were not intending to downplay our results. Rather we wanted to make sure that we emphasized that one study like this is a first step in this process. We need to open the door to future research and unfortunately there is a tendency for a single study to be taken by the public as the “solution” to a problem that is just being understood.

That said, we have completely revised the organization of the discussion section and reworked this concluding paragraph to emphasize the fact that this is a first step towards a greater understanding of the problem.

See also attached figure. I would be happy to share my illustrator-file of that figure should the authors be interested in precisely redrawing the abiotic drivers using the original file next to the divergence figure.

-Christian Rasmussen

Appendix B

To the editors of the Proceedings of the Royal Society B,

We would like to resubmit our manuscript, “An Early Burst in brachiopod evolution corresponding with significant climatic shifts during the Great Ordovician Biodiversification Event,” for consideration as a research article. This paper uses fossilized birth-death modeling to estimate the rates of cladogenesis and the timing of diversification of strophomenoid brachiopods during the Great Ordovician Biodiversification Event, one of the largest evolutionary radiations in Earth’s history. We have addressed all of the specific recommendations from Reviewer 2, and we have expanded our references in our discussion at the request of Reviewer 1. The specific references added are listed at the end of this letter.

Thank you very much for your consideration.

Sincerely,

Curtis R Congreve, Peter J. Wagner, and Mark E. Patzkowsky

- Bassett, M. G., et al. 2002. Brachiopods: Cambrian-Tremadoc precursors to Ordovician radiation events. *Geol. Soc. Lond., Spec. Publ.* 194:13 - 23. (10.1144/GSL.SP.2002.194.01.02)
- Colmenar, J. and C. M. Ø. Rasmussen. 2018. A Gondwanan perspective on the Ordovician Radiation constrains its temporal duration and suggests first wave of speciation, fuelled by Cambrian clades. *Lethaia* 51:286-295. (10.1111/let.12238)
- Miller, A. I. and S. R. Connolly. 2001. Substrate affinities of higher taxa and the Ordovician Radiation. *Paleobiol.* 27:768 – 778. (10.1666/0094-8373(2001)027<0768:SAOHTA>2.0.CO;2)
- Zhan, R., et al. 2013. The earliest known strophomenoids (Brachiopoda) from early Middle Ordovician rocks Of South China. *Palaeont.* 56:1121-1148. (10.1111/pala.12039)
- Zhan, R.-B. and D. Harper. 2006. Biotic diachroneity during the Ordovician radiation: evidence from south China. *Lethaia* 39:211 - 226. (10.1080/00241160600799770)
- Zhan, R. B., et al. 2006. β -diversity fluctuations in Early–Mid Ordovician brachiopod communities of South China. *Geological Journal* 41:271-288. (10.1002/gj.1040)